# Food, Energy, and Water Nexus at Household Level: Do Sustainable Household Consumption Practices Promote Cleaner Environment?

**DOI:** 10.3390/ijerph191912945

**Published:** 2022-10-10

**Authors:** Pomi Shahbaz, Shamsheer ul Haq, Azhar Abbas, Abdus Samie, Ismet Boz, Salim Bagadeem, Ziyue Yu, Zhihui Li

**Affiliations:** 1Department of Agricultural Economics, Ondokuz Mayıs University, Samsun 55139, Turkey; 2Department of Economics, Division of Management and Administrative Science, University of Education, Lahore 54770, Pakistan; 3Institute of Agricultural and Resource Economics, University of Agriculture, Faisalabad 38040, Pakistan; 4Faculty of Business Administration, Arab Open University, Riyadh 11681, Saudi Arabia; 5Institute of Geographic Sciences and Natural Resources Research, Chinese Academy of Sciences, Beijing 100101, China; 6Department of Architecture and Built Environment, The University of Nottingham Ningbo China, Ningbo 315100, China

**Keywords:** energy sustainability, environmental conservation, ecosystem services, GHG emission, consumer behavior

## Abstract

Governments around the globe are trying to find sustainable solutions for lessening pressure on natural resources and reducing carbon emissions. Daily household consumption of food, energy, and water has an impact on stocks of natural resources, environmental quality, and climate change. Households have significant potential for increasing conservation actions for efficient use of natural resources and greenhouse gas emissions. Households could contribute to a clean and healthy environment by adopting sustainable household practices through lower per capita consumption and carbon emissions. This study explored the role of different sustainable household consumption practices in promoting a clean environment as well as the factors affecting the adoption of these practices in Pakistan. Factor analysis and an ordered probit model were used to analyze the data from 1424 participants chosen through a multistage random sampling technique. The factor analysis identified 35 sustainable household practices for sustainable consumption. These 35 practices were grouped into the underlying factors of “Food” (14 items), “Energy” (12 items), and “Water” (9 items). The results from the econometric model showed a significant relationship between gender, education, residential area, family size, and income and the adoption of sustainable household consumption practices. Statistically, higher levels of reported sustainable consumption practices were apparent among females, households living in urban areas, more educated people, individuals of large family sizes, and more affluent households. Therefore, public policies for taking care of the environment need to put households at the center while at the same time promoting mass uptake of sustainable consumption practices related to food, energy, and water. In addition, the sector-specific policies also need to be augmented through focus on household-level consumption and production dynamics for achieving the UN’s SDGs.

## 1. Introduction

The grand challenges of the 21st century are sustainability challenges, and the world is struggling to contain current anxieties about climate change, environmental degradation, and economic instability, mainly due to unsustainable consumption of goods and services [1]. Household consumption of goods and services holds a significant share in global resource consumption as well as in associated environmental impacts [2]. Households consume a substantial amount of food, energy, and water resources either directly (preparing food and doing laundry) or indirectly (consequences of food and energy production on the land and water) [3]. Moreover, households account for almost three-fourths of the global greenhouse emissions worldwide and the average per capita carbon footprint generation is 6 tCO2eq [4,5]. Households in developed countries have greater average per-person carbon footprints than the rest of the world. Thus, household consumption patterns and behavior have an impact on stocks of natural resources, environmental quality, and climate change [6]. Households could cut carbon emissions by up to 15 billion tons by 2060 simply by eliminating meat from their diets or avoiding flying [7]. This indicates that households have significant potential for increasing conservation actions for efficient use of natural resources and greenhouse gas emissions [8].

Thus, by engaging in long-term sustainable behavior, household consumers may significantly contribute to a clean environment and sustainable development. The consumption of food, energy, and water by households is a part of their culture, habits, and daily routine. Conventional approaches only seldom take into account how various households utilize this information in their day-to-day lives. For instance, individuals often do not consider energy use while they are driving to work, cooking, bathing, lighting their rooms, or heating and cooling their homes. Reid et al. [9] and Lane and Gorman-Murray [10] contend that when discussing sustainability, the household should be highlighted as an organizational unit. This is so that the situated meanings of consuming behaviors may be revealed, which happens in homes. Governments around the world are trying to find viable solutions for reducing carbon emissions and pollution, one of the primary sources of which is domestic household consumption.

Therefore, household consumption behavior is part and parcel of global climate and environmental policies. The sustainability of our planet is dependent on the willingness of individual households to modify their long-term food, energy, and water consumption behaviors [11]. The topic of behavioral change has often been sidelined in debates about global climate policy due to the focus on technology and financial incentives. Therefore, in addition to focusing on supply-side mitigation via technology or legislation, we must put far more emphasis on demand-side solutions, i.e., reducing consumption. Additionally, we must think of lifestyles as objectives for policy (and modeling initiatives) rather than as voluntary contributions made by people. In fact, if the world is to meet the goals of the Paris Agreement, which include reducing carbon dioxide emissions from 40 gigatons in 2020 to 5 gigatons in 2050 and reaching “net zero” by 2100, emissions must be cut much more sharply. This basically means that emissions must permanently decline by 50% every 10 years until the middle of the century, at which point they must surpass a certain barrier and be further reduced by instituting so-called “negative emissions” policies through the end of the century [12]. Then, when it comes to emission inventories and mitigation plans, we must take a more balanced approach to household consumption practices.

Discussions on how society will transition to a low-carbon and green economy center on the need to urge individuals to modify their consumption practices and behavior to support a clean environment. Promoting responsible consumption and production among households is one of the key strategies for attaining the 12th Sustainable Development Goal of the UN. This objective stems from the notion that lifestyles, demands, and desires that use a lot of natural resources are responsible for the current environmental catastrophes [13,14]. Households should utilize food, energy, and water in ways that assist in improving environmental quality because the growing commercialization of human activities has aggravated environmental anxieties [15].

The recent emphasis on household consumption patterns is a result of the significance of household emissions to climate policy. Understanding the factors that influence sustainable consumption behavior for environmentally friendly consumption is necessary for an efficient consumer-oriented climate strategy. In the years to come, household environmental pressures are anticipated to intensify if further government actions are not taken. Governments are increasingly focusing on the study of environmental policy from the demand side, but it is still challenging to build development plans that promote more environmentally friendly ways of living and consuming food, energy, and water. Therefore, household consumers are increasingly being highlighted as a key area requiring attention for sustainable development. 

Since it is known that households influence sustainable consumption, studies on consumer practices and sustainable lifestyles have been undertaken for decades. Although numerous studies on sustainable lifestyles and household consumption practices have been conducted [16,17,18,19,20,21,22], the majority of these studies were carried out in developed countries, ignoring the significant population that lives in developing countries. Additionally, the bulk of this research concentrated on a single area of consumption: food, water, or energy. Another drawback of earlier research in this area was the preponderance of studies that concentrated on urban households when examining household consumption practices. This research aims to fill the aforementioned gaps in the literature by examining sustainable consumption strategies of food, energy, and water together in a developing nation. Thus, the first objective of this study was identifying sustainable consumption practices adopted by the households for a clean environment. The second objective of this study was to shed light on the factors which shape the sustainable consumption behavior of households for a clean environment. The first hypothesis was that household dynamics affect the adoption of sustainable household consumption patterns. Another hypothesis under this objective was that households’ awareness of climate change also affects the sustainable household consumption practices. The other objective of this research was to determine whether sustainable consumption behavior leads to lower waste generation in different households.

The research has significant policy ramifications because we need to learn more about what could persuade households to embrace low-carbon technology and ways of life, regardless of whether changes in food, energy, and water consumption are proactive or reactive. Additionally, it is essential to comprehend the barriers to and motivating factors for consumer behavior improvements, especially in light of how urgently we want worldwide decarburization approaches.

The remainder of this article is organized as follows: The study’s materials and methods are discussed in Section 2 of the paper. The study’s results and discussion are included in Section 3. In Section 4, the conclusions and policy recommendations of the study are given, along with the limitations of the study. 

## 2. Materials and Methods

### 2.1. Study Design and Data Collection Instrument

A cross-sectional study design was developed for the current study. To accomplish the data collection task, we developed a questionnaire, which consisted of four sections to make the data collection convenient. The first section contained questions regarding the household dynamics of the participants. These were the age, education level, gender, main occupation, residential area, income level, family size, society membership, internet access, and family income level of participants. The second section was about participants’ awareness of and giving importance to climate change. The third section was about sustainable household practices, which focused on the consumption of food, energy, and water, contributing to sustainable household capabilities. The last section was about the waste generation frequency of different important wastes at the household level. The validity of the questionnaire was confirmed by conducting a pilot survey, and the ambiguities were removed before the final survey. The majority of the changes made in the data collection instrument were related to household consumption practices because the practices included in the pilot survey questionnaire were those taken from previous work in this field. Household consumption practices were modified to match the ground realities before the beginning of the final survey. Additionally, since female participants were reluctant to offer data to male enumerators in pilot survey, female enumerators were also added to the data collection team to increase the participation female respondents in the final survey. The questionnaire was finalized and developed in the English language. The face-to-face interview was conducted in the local language.

### 2.2. Study Area and Sampling Procedure

The suitable sample size for the current study was derived through the following Krejcie and Morgan’s [23] formula:$$n=\frac{X^{2}*N*P*\left(1-P\right)}{d^{2}*\left(N-1\right)+(X^{2}*P*\left(1-P\right)},$$
where *n* is equal to the sample size required to accomplish the current study. $X^{2}$ indicates the value of chi-square for a specified confidence level at 1 degree of freedom (3.841). The letter “*N*” denotes the population, which was the number of households in Pakistan; “*P*” is the population share that was assumed to be 0.50 here; and “*d*” denotes the sampling error, which was assumed to be 2.6% for the current study. This resulted in a total required sample size of 1421. The sample size of 1421 was distributed across the smallest residency units both in urban and rural areas through a multistage random sampling technique. 

Punjab was chosen to be a representative province of Pakistan because it is the second largest province in terms of land area and has the most people of any province, with more than 109 million. Moreover, there are 535.63 people per square kilometer in the province, which is a lot more than in other provinces in the country. Residents of this province have suffered greatly as a result of recent natural disasters caused by climate change, and no province better represents Pakistan than Punjab. Therefore, the current study was planned to be conducted in Punjab Province.

Punjab Province consists of thirty-six districts, of which five were selected randomly. Each district consists of several towns (local = tehsil) and two towns from each district were chosen randomly. Each town further consists of many union councils, and two union councils were selected from each selected town. Each union council is made up of many villages in rural areas and neighborhoods (*mohalla* in local language) in cities. The next step was to select the villages and neighborhoods from each union council, and four villages and neighborhoods from each union council were selected on a random basis. In the end, the sample size of 1421 was proportionately distributed across a total of eighty villages and neighborhoods, which resulted in a number equal to 17.76 (=1421/80) for each village and neighborhood for the survey. By rounding the value from 17.76 to 18, the data from 1440 households were collected. After data collection, 16 incomplete samples were removed, and a total of 1424 samples were considered for further analysis in this study.

### 2.3. Measuring Sustainable Household Consumption Index

Taking into account previous studies, the culture of the country, the researchers’ own observations, and the opinions of experts, a total of 56 sustainable household consumption practices were identified for a clean and green environment. Participants’ responses about the adoption of any sustainable household consumption practice were recorded on a 5-point Likert scale. This scale is best for collecting data from a large number of survey participants, and it also provides highly reliable estimates [24]. The content validity was confirmed by five experts having subject matter as well as cultural knowledge of the study area. The frequency 5-point Likert scale was used, which included “never,” “rarely”, “sometimes”, “often”, and “always”. The response of a participant to a practice describes how often a household adopts a practice or action when one performs a certain activity such as cooking, eating, washing, and using energy. After data collection, household responses were coded according to the contribution toward a clean environment. For example, if a household chooses to always switch off lights in unoccupied rooms, it indicates the household’s highest contribution towards sustainability and a clean environment. Therefore, this kind of response was coded as 5. Similarly, “often” was coded as 4, “sometimes” was coded as 3, “rarely” was coded as 2, and “never” was coded as 1. 

Afterward, the scale was factor-analyzed in order to determine if the data might be utilized to identify common factors. The extraction and rotation methods were principal component analysis and varimax with Kaiser normalization. The factor analysis identified 35 sustainable household practices for sustainable consumption. These 35 statements were grouped into the underlying factors of “Food” (14 items), “Energy” (12 items), and “Water” (9 items). Factor analysis is commonly used to group the items for Likert scale statements [25] (ul Haq et al., 2022). Each of the three derived components had Cronbach alpha values greater than the cutoff (0.70), indicating a high degree of internal reliability. The Cronbach alpha value greater than 0.70 for all constructs implies internal consistency among construct elements [26].

The sustainable household consumption index was developed with a total of 35 remaining items from the factor analysis procedure regarding their Likert scale responses (the lowest score of sustainable household consumption index was 35 = (35 × 1), and the highest score of sustainable household consumption index was 175 = (35 × 5)). Thus, the sustainable household consumption index varies between 35 and 175. Considering the frequency distribution of the sustainable household consumption index, the respondents were divided into three groups of sustainable households. The low-sustainability households were those having a sustainable household consumption index score of less than 100, which were 382 in number and comprised 26.83% of the total sample size. The medium-sustainability households were those securing a sustainable household consumption index score in the range of 101–125, and they were 643 in total, making up 45.15% of the total sample size. Participants having sustainable household consumption index scores greater than 125 were grouped as “high-sustainability households” and there were 399 such households, representing 28.02% of the total sample size.

These groups were used as the dependent variable of the ordered probit model that was used to achieve the second objective of this study. The current study is highly dependent on Likert scale responses, which generate the data in the ordinal or ordered form. Consequently, the dependent variable was developed into an ordered variable because the central idea is that the fundamental response is latent, representing the performance of households in terms of adoption of sustainable consumption practices. Daykin and Moffatt [27] described how an ordered probit model deals consistently with ordinal data. Clark et al. [28] and Haq et al. [25] also used the ordered probit model to explore determinants of life satisfaction and sustainable perception, while data responses were recorded on a Likert scale.

### 2.4. Explanatory Variables

#### 2.4.1. Household Dynamics

Household dynamics are crucial to study consumption behavior of different households. The explanatory variables for this study were considered by taking into account the previous studies [29,30,31,32,33] in the field as well as keeping in view the ground realities of the study area. These variables included age, education, gender, family size, monthly family income, main occupation, residential area, society membership, and internet access. The age and education of respondents were measured in years and directly entered into the model as continuous variables. Family size was recorded in numbers and household income was enquired about in Pakistani rupees (PKR). The households were categorized into low-, medium-, and high-income households. Households having a monthly income of less than PKR 50,000 were categorized as low-income households; households having an income of between PKR 50,000 and 100,000 were grouped into the medium income category; and households having an income of greater than PKR 100,000 were called high-income households. The other remaining explanatory variables, such as gender, residential area, internet access, and main occupation were recorded in the form of dummy variables. These variables were coded as gender (1 for male and 0 for female), residential area (1 for urban and 0 for rural), the main source of income (1 for non-agriculture and 0 for agriculture), society membership (1 for yes and 0 for no), and internet access (1 for yes and 0 for no).

#### 2.4.2. Climate Change Awareness, Importance and Knowledge of Sustainable Household Consumption Practices

Climate change awareness is the key to sustainable consumption and clean environmental behavior. Several studies [29,34,35] emphasized the significance of environmental and climate change knowledge as a primary driver of clean and green consumption behavior. In this study, the climate change awareness of households was measured with three questions: 1. “Awareness level related to the causes of climate change”; 2. “Awareness level related to the impacts of climate change on the planet”; and 3. “Awareness level related to the role of household consumption on climate change”. All these questions were recorded on a 10-point Likert scale (1 for very low and 10 for very high). The mean scores of all three questions were used as an explanatory variable in the econometric model.

Similarly, considering climate change as a crucial issue for the local area and country could lead to the adoption of sustainable household consumption practices. Climate change’s importance was also measured by three questions on a 10-point Likert scale, from not very important to very important: 1. “Climate change a crucial problem for the country” and 2. “Climate change a crucial problem for the local community (district/province)” and 3. “Climate changes an important issue for you”. The mean of these three statements was used as a variable in the model to explain the adoption of sustainable consumption practices of households.

The knowledge of sustainable household practices was measured by two questions on a 10-point Likert scale: 1. “Knowledge of sustainable household practices” and 2. “Knowledge of unsustainable household practices on human wellbeing”.

### 2.5. Hypothesizing the Effect of Explanatory Variables on Dependent Variable

Figure 1 shows the expected effects of all explanatory variables on sustainable household consumption in the light of previous work as well as due to cultural context. Young households are expected to be more aware of sustainable household practices and their impacts on climate change. Moreover, younger households are always more enthusiastic about learning and are open to learning new things. On the other hand, older households are always hesitant to change their behaviors and present practices. Thus, age is assumed to negatively influence the adoption of sustainable consumption practices.

Education is expected to correlate positively with the adoption of sustainable household consumption practices. Educated people are expected to have higher awareness and information about sustainable consumption practices of different resources at home, such as food, energy, and water. Educated people may know very well the impact of unsustainable consumption practices on the environment, and this helps educated people adopt sustainable practices to play their role in a clean and green environment. Rezvanfar et al. [36] also reported that education positively affects the adoption of sustainable practices. Family size was assumed to be both positively and negatively associated with the adoption of sustainable household practices. Yates and Evans [18] noted that a smaller family size has an impact on sustainability and that resources are utilized more sustainably when individuals live together. On the other hand, household resources can also be wasted due to the lavish consumption of households if all households do not behave responsibly.

The female gender is hypothesized to positively correlate with the adoption of sustainable consumption practices. The reason is that a number of studies [30,37] have identified females as more pro-environment and climate change sensitive than males all over the world. Similarly, a household living in an urban area is expected to effect sustainable consumption behavior positively. The reason is that we expect higher sustainability awareness and education levels among urban people in the country due to the main focus of public policies on cities. A number of studies specifically focus on sustainable cities, and rarely is a study found on sustainable rural areas or villages.

A person having a non-agricultural occupation may have a positive impact on the adoption of sustainable household consumption practices. Income can play an important role in the adoption of sustainable household practices because rich and wealthy households are more likely to have efficient appliances and modern technologies at home. Having internet access and society membership is also supposed to have a positive impact on the adoption of sustainable household practices. In a similar way, this study assumes that knowing about sustainable practices and being aware of climate change will have a positive effect on the adoption of sustainable household consumption practices. Prior studies also signify the importance of climate change awareness and knowledge of sustainable household consumption practices for their adoption.

### 2.6. Ordered Probit Model

The dependent variable was coded as 0 for low-sustainability households, 1 for medium-sustainability households, and 2 for high-sustainability households in the ordered probit model, which is expressed as:y* = *β*′*x_i_* + *ε*,*ε* ∼ *N* (0, 1)
y = 0    if y* ≤ 0
y = 1    if 0 < y* ≤ *μ*_1_
y = 2    if *μ*_1_< y* ≤ *μ*_2_,
where y* is the dependent variable of the model, which describes the probability of a respondent belonging to a sustainable household category. *β*’ is the vector of coefficients; *x_i_* is the vector of the independent variables; and ε denotes the vector of normally distributed error terms [0, 1]. Finally, y depicts the observed dependent variable as the probability of respondents who belong to a high-sustainability household, and *μ* shows the cutoff points, which present the level of inclination of respondents to have a sustainable household. The marginal effect was measured by using the formula proposed by Chen et al. [38], and it explains how much each dependent variable increases or decreases the probability of a respondent being in each of the three categories of the dependent variable.
$$\frac{\partial P\left(y_{i}=j\right)}{\partial x_{k}}=\left[\Phi [\mu_{j-1}-\overset{k}{\underset{k-1}{\sum }}\beta_{k}x_{k}\right]-\Phi \left[\mu_{j}-\overset{k}{\underset{k-1}{\sum }}\beta_{k}x_{k}\right]\beta_{k}$$

$\partial P/\partial x_{k}$ is a partial derivative of probability with respect to the independent variable. The positive value of the marginal effect describes that the probability of a respondent choosing a specific category increases with *x_k_* and vice versa. The sum of the marginal effects should be zero because the responses are exclusive and thus cancel each other out [39].

## 3. Results and Discussion 

### 3.1. Sample Background

The background information about the households under study always plays a key role in understanding their short- and long-term behavior and decision-making. Household dynamics affect different decisions both positively and negatively. The age of households is one of the most important factors that has the potential to affect the consumption practices of the household. A young person can also help them better by helping put their behavior and problems in context. The sampled households were almost evenly distributed between the young and old categories. In Pakistan, as in other developing countries, it is hard for women to take part in surveys because of their culture. So that a study is a true reflection of society, researchers in developing countries such as Pakistan must always make an extra effort to obtain more female respondents to take part in surveys. As a result of these special measures, more than two-fifths of the participants in this study were females (Table 1). The education level in Pakistan is not very high among households as the overall literacy rate of the country is low [40]. This may be why more than two-fifths of the study’s participants fell into the low-education group.

More than half of the participants in this survey live in rural areas. The reason may be that despite the rapid urbanization in the country, more than two-thirds of the total population in Pakistan still resides in rural areas of the country [41]. Although the share of agriculture in the economy of the country is declining each year, agriculture is still the mainstay of living for more than one-third of the residents of the country. The survey results also showed the same as almost one-third of the participants pointed out agriculture as their main source of income. Large family sizes are common in Pakistan due to joint family systems prevailing in the country, and that might be the reason that more than half of the households belonged to large families [42]. Only less than one-fifth of the participating households had any membership in a society. This may be because of the lower education level of the participants and the unavailability of the membership of any society in the country. Internet services are expanding rapidly, and the country has more than 100 million internet users [43]. More than half of the participants were using the internet to obtain different types of information. More than half of the study participants belonged to the low-income category. This may be because the income level in Pakistan is not very high and Pakistan is placed in the lower middle income category by the World Bank.

The results related to climate change awareness showed that more participants were aware of the causes and impacts of climate change than the participants who had knowledge of the role of sustainable household consumption on the Earth’s climate. Overall, the climate change awareness and importance level of the participants were very satisfactory. A majority of the participants also consider climate change an important issue domestically and nationally. Moreover, a large majority of the households had lower knowledge of sustainable household consumption practices and their effect on human wellbeing.

### 3.2. Identifying the Sustainable Household Practices for Sustainable Consumption

The factor analysis identified 35 sustainable household practices for sustainable consumption in three factors labeled as food, energy, and water consumption factors, according to the items included in each factor. It would be difficult to evaluate correlation and different calculations between each of these factors separately and their associations with the chosen demographics. Additionally, it could greatly inflate the experimental error (alpha level). The scale was factor-analyzed in order to determine if the data might be utilized to identify common factors. The rotated component matrix for sustainable household consumption practices is presented in Table 2. All three factors accounted for 58.23% of the variance, and the food factor alone explained almost half (27.39%) of the total variance. This was followed by energy and water factors, with 18.56% and 12.56% variance explanations, respectively. Moreover, the factor scores for household sustainable consumption factors were calculated as mean = 3.61 (SD = 0.69) for Factor 1—Food factors, mean = 3.92 (SD = 0.72) for Factor 2—Energy factors, and mean = 3.12 (SD = 0.91) for Factor 3—Water factors.

Analysis of household Likert scale responses revealed food-related factors are the most important for sustainable household consumption. Among the practices covered by this factor, households placed a higher value on sustainable consumption, such as cooking according to daily needs, storing extra food for reusing, avoiding purchasing food items already in the home, implementing a FIFO strategy for kitchen management, preferring domestic food, eating organic, not using disposable plates, avoiding drinks packed in aerosol containers, etc. For sustainable use of energy resources, households prioritized switching off lights in unoccupied rooms, using energy-efficient electric appliances, using fewer lights, washing laundry by hand and drying in the sun, watching TV together, less ironing of clothes, using regular water for bathing and washing clothes, etc. Water-related components included avoiding leaving the tap running while washing dishes, cleaning teeth, and soaping the body and face, shortening bath time, and avoiding water tank overflow, among other things. 

### 3.3. Determinates of Sustainable Household Consumption Practices

Sustainable household consumption is critical for lowering greenhouse gas emissions and effectively and efficiently using resources for a clean environment. Sustainable consumption behavior is partly contingent on household dynamics and individuals’ climate change awareness. Household dynamics are important to understand the different levels consumption practices of the participants. The ordered probit model was utilized to determine the factors affecting the level of sustainable household consumption practices. The coefficient and marginal effects of the ordered probit model are shown in Table 3. With a log likelihood ratio of chi-square value of 269.123 and a probability of chi-square of less than 1%, the overall model was significant. Out of the thirteen explanatory variables entered in the econometric model, nine (gender, education, residential area, family size, income, climate change awareness, climate change impotence, and sustainable consumption knowledge) were found to be significantly affecting the adoption of sustainable consumption practices of the households. 

Gender is the major factor out of all the other factors that affects sustainable consumption practices. Gender was negatively associated with the level of sustainable consumption behavior of households. Male households are 5.2% less likely than female households to fall into the high-sustainability consumption category. Similarly, male households were 0.2% points and 5% points more likely to belong to the low- and high-sustainability consumption categories, respectively. This may be because males and females tend to have different consumption practices because of the difference in their upbringing and socialization. The results are also consistent with the previous studies [30,37,44], which describe that females are more concerned about the environment than males. Dos Muchangos and Vaughter [45] reported that compared to males, females have comparatively more sustainable lifestyles. Moreover, they also stated that males’ consumption patterns have larger environmental footprints than females’.

Household education was found to be positively correlated with the adoption of sustainable consumption practices. If a household’s education level goes up by one year, the chances of being in the low- and medium-sustainability consumption groups go down by 0.04% and 1.2%, respectively. On the other hand, a one-year increase in the education of the households increases their probability of being in the high-sustainability consumption group by 1.6% points. Figueroa-García et al. [31] also reported that the education level of households positively influences the sustainable consumption of households. The results of the study also agree with what Kollmuss et al. [46] and Bamberg and Moser [47] found about the relationship between sustainable consumption and the education level of the household. 

The residential area of households is also one of the important determinants of sustainable consumption of households, and the difference in everyday life and sustainable consumption behavior between urban and rural areas is obvious. Residential areas were found to be positively related to sustainable household consumption. Urban households are 2.9% points more likely to belong to the high-sustainability consumption group. On the other hand, urban households are 0.09% points and 2% points less likely to belong to the low- and medium-sustainability consumption groups. The reason may be that households living in urban areas have to purchase food and water in Pakistan compared to rural areas where people grow and manage this themselves, which may lead to these resources’ unsustainability. Another difference could be that people in urban homes may be more aware of sustainability than people in rural homes, since public policies all over the world focus a lot on urban sustainability. 

In high-income countries, housing has a big impact on the demand for natural resources and the acceleration of economic development [48]. Family size of households was also found to be positively influencing the sustainable consumer behavior of households. A one-member increase in the family size of the households increases their probability of being in the high-sustainability consumption group by 17% points. On the other hand, a one-member increase in households’ family size reduces the likelihood of being in the low- and medium-sustainability consumption groups by 7% points and 10.1% points, respectively. Yates and Evans [18] also reported that resources are used more sustainably when people live together, and that decreasing family size has repercussions for sustainability.

Household income is also one of the most important determinants of sustainable household consumption. When compared to low-income households, medium-income households are 1% and 2.4% less likely to belong to low and moderately sustainable consumption groups, respectively. In the same way, high-income households are 1.8% points and 4.5% points less likely than low-income households to be in the low and moderately sustainable consumption groups. Medium- and high-income households are, respectively, 3.4% points and 6.1% points more likely to belong to the highly sustainable consumption group. This shows that household income is positively affecting the sustainable consumption of households, and sustainable consumption gradually increases as the income level of the household increases. Wang et al. [49] also reported that household income positively and significantly affects the sustainable consumption of households.

Climate change awareness and considering climate change an important issue are critical to the adoption of sustainable consumption practices because environmental concern is shown to strongly influence behavioral intention, which further mediates sustainable consumption behavior. The results of an ordered probit model showed that climate change awareness and considering climate change an important issue positively affect the sustainable household consumption. A household having climate change awareness is 4.8% points more likely to belong to the high-sustainability consumption category. On the other hand, a household having climate change awareness is 2.2% points and 2.6% points less likely to be in the low- and medium-sustainability consumption categories, respectively. Similarly, a household considering climate change an important issue is 3.6% points and 8.7% points less likely to be in the low- and medium-sustainability consumption category, respectively. A household having climate change awareness is 12.3% points more likely to belong to the high-sustainability consumption category. The study corroborates prior studies [29,49,50,51] showing that environmental knowledge and awareness positively influence the sustainable consumption of households. 

### 3.4. Waste Production Comparison of Different Sustainable Consumption Groups

Pakistan produces copious amounts of waste every year, and Figure 2 shows the waste generation of the different households divided into low-, medium-, and high-sustainability households based on the adoption of sustainable consumption practices in their daily lives. The results showed that sustainable consumption of food, energy, and waste has an impact on the frequency of waste production in homes, and households that tend to be more sustainable consumers also produce different types and less waste. Kitchen waste is the most common type of waste, while e-waste is the least common type of waste generated in all three sustainable consumption categories. This was to be expected because kitchen items are used more often than other waste items on the list. As a result, these kitchen items are thrown away more often. Ilyas et al. [52] also stated that food and kitchen waste are the most common types of trash in Pakistan’s big cities. 

The number of households producing kitchen waste on a daily basis in the low- and medium-sustainability groups was higher than the number of households in the high-sustainability group. Most of the waste generated at home consists of food scraps such as fruit peels, vegetables, and spoiled products [53]. Noufal et al. [54] also reported in their study that food organic waste constitutes the largest share for households. Similarly, a higher number of households in the high-sustainability household category produce glass waste occasionally and rarely as compared to the number of households in the low- and medium-sustainability household categories. More low- and medium-sustainability households than high-sustainability households pointed out that they make plastic waste every day and every week. The number of households in the high-sustainability household group mentioned waste generation on an occasional and rare basis compared to the number of households in the low- and medium-sustainability household groups. In addition, given a higher risk of climatic extremes, the need for integration of food production and natural resource conservation such as water and energy are highlighted as a main agenda to ensure food security and climate change adaptation and resilience building especially among farming community in the region [55,56].

## 4. Conclusions

Sustainable household consumption practices are critical to transit towards a low-carbon economy and a clean environment. Therefore, public policies for taking care of the environment need to put households at the center. The study results highlight the significance of public policies based on daily household consumption practices rather than sector-specific policies. Second, the results of the study suggest that the government should take steps to recognize how households with different dynamics adopt sustainable consumption practices. Third, the study’s results indicate the necessity for government initiatives that are sensitive to households’ knowledge and awareness of everyday sustainable consumption patterns.

The factor analysis identified 35 sustainable household practices for sustainable consumption. These 35 practices were grouped into the underlying factors of “Food” (14 items), “Energy” (12 items), and “Water” (9 items). The sustainable household consumption index was developed by utilizing the scores of these 35 Likert scale statements (the lowest score of the sustainable household consumption index was 35 = (35 × 1), and the highest score of the sustainable household consumption index was 175 = (35 × 5). Based on the sustainable household consumption index scores, households were categorized into three main groups, namely low-, medium-, and high-sustainability households. The low-sustainability households were those with a consumption index score of less than 100, which comprised 26.83% of the total sample size. The medium-sustainability households were those securing a sustainable household consumption index score in the range of 101–125, and made up 45.15% of the total sample size. Households having sustainable consumption index scores greater than 125 were named “high-sustainability households” and they represent 28.02% of the total sample size.

### Policy Recommendations

The study’s results also show that if public policies are to be effective in addressing everyday unsustainable consumption patterns, they must also take into account the more complicated household dynamics that affect consumption. Household dynamics have a significant role when considering human activities in relation to the routines that enable sustainable household consumption behavior. An econometric model result showed a significant relationship between gender, education, residential area, family size, and income and the adoption of sustainable household consumption practices. Statistically, higher levels of reported sustainable consumption practices were apparent among females, households living in urban areas, more educated people, individuals of large family sizes, and more affluent households. In light of the above results of factors affecting sustainable household consumption practices, public policy should focus on shifting the daily household practices of the less affluent and male households toward more sustainable consumption for a cleaner environment. Likewise, public policies must acknowledge that women and large households are already doing the work for a clean environment and have the least ability to improve household consumption practices. Households residing in urban homes exhibited greater adoption of sustainable household practices for low carbon emissions and a clean environment. Moreover, the situated meanings that define environmental challenges within a household’s social interactions and regular routines make the cultural values of families important as well. However, the changing cultural context alone is not sufficient to improve daily consumption for more sustainable households. Climate change awareness of households also plays a critical role, as shown by its significant relationship with sustainable consumption behavior.

In the context of the challenges of sustainable consumption practices, the cross-sectional survey approach limited our research. The other limitation of the study is the lack of consideration of waste disposal practices by the households, which are an important part of sustainable household consumption practices and necessary for a clean environment. Future research can go further than what was done in this study by looking at different ways to deal with home waste, along with the sustainable consumption practices considered in this study, by collecting longitudinal data in other developing countries using different sampling methods.

## Figures and Tables

**Figure 1 ijerph-19-12945-f001:**
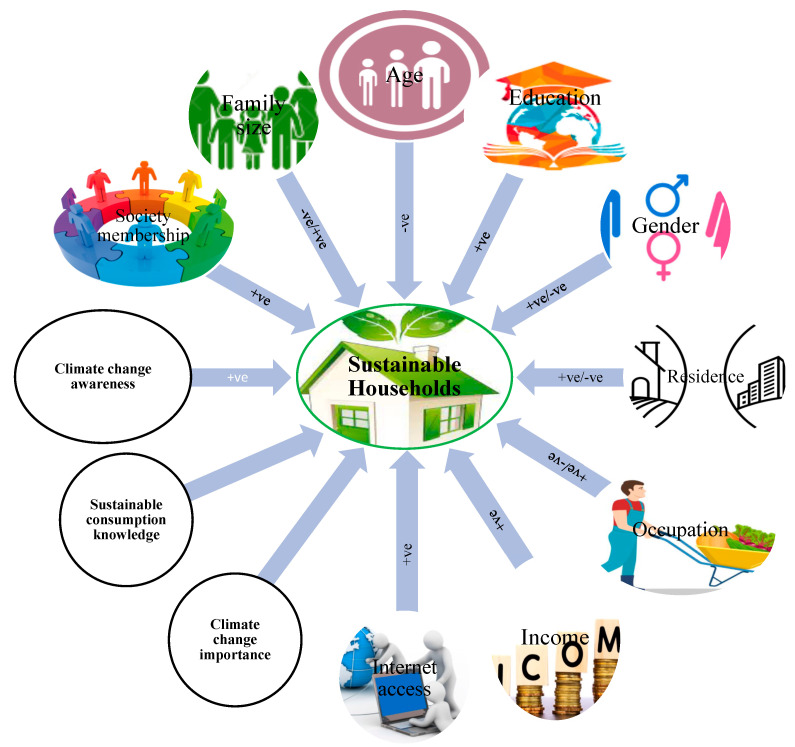
Expected relationship of explanatory variables with sustainable household practices.

**Figure 2 ijerph-19-12945-f002:**
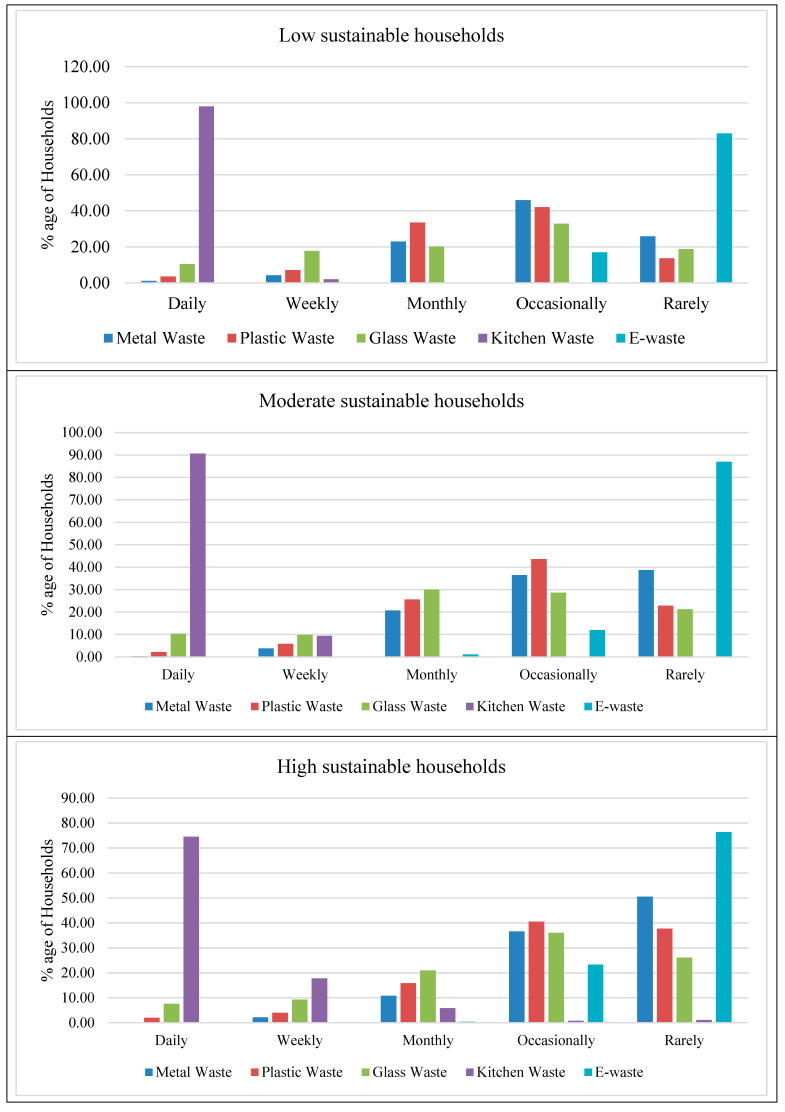
Frequency of waster production of different sustainable consumption groups.

**Table 1 ijerph-19-12945-t001:** Household dynamics, sustainable household consumption knowledge, and climate change awareness and importance indicators.

Sample Background	Number	Percentage
Age (Years) ^a^		
Young	687	48.2
Old	737	51.8
Gender		
Female	570	40.1
Male	860	59.9
Education (Years) ^a^		
Low	634	44.6
High	790	55.4
Residential area		
Urban	648	45.5
Rural	776	54.5
Main source of income		
Agriculture	464	32.6
Non-agriculture	960	67.4
Family size (persons) ^a^		
Small	704	49.5
Large	720	50.6
Society membership		
No	1176	82.6
Yes	248	17.4
Internet access		
No	648	45.5
Yes	776	54.5
Monthly income (PKR)		
Low (<50,000)	768	53.9
Medium (50,000–100,000)	521	36.6
High (>100,000)	135	9.5
Climate change awareness and importance	Mode	
Awareness level related to the causes of climate change ^b^	7	
Awareness level related to the impacts of climate change on Earth ^b^	8	
Awareness level related to role of household consumption on climate ^b^	5	
Climate change awareness (mean)	6.81	
Climate change a crucial problem for you ^b^	7	
Climate change a crucial problem for your district/province ^b^	8	
Climate change a crucial problem for the country ^b^	9	
Climate change importance (mean)	7.36	
Knowledge of effect of sustainable household practices on human wellbeing ^b^	4	
Knowledge of sustainable household practices ^b^	5	
Sustainable household consumption knowledge	4.45	

^a^ = the sample was separated into two groups by taking average of the sample as the cutoff point. ^b^ = the responses were recorded by using a 10-point Likert scale.

**Table 2 ijerph-19-12945-t002:** Factor analysis of sustainable household consumption practices.

Factors and Related Items		Components		
A. Food		1	2	3
1	Cook according to daily needs	0.795		
2	Freeze extra food properly for reuse	0.783		
3	Avoid buying food items already present	0.764		
4	Management of kitchen with first in first out (FIFO) strategy	0.759		
5	Include only those ingredients in food which one likes	0.726		
6	Buy local and domestic food produce	0.689		
7	Buy organic produce	0.674		
8	Avoid using disposable tissue papers	0.660		
9	Consume food products without packaging	0.659		
10	Avoid using food/drinking products in aerosol containers (tin)	0.643		
11	Avoid food products in disposable plastic/ bottles	0.637		
12	Use clothes instead of paper to clean the kitchen	0.629		
13	Use homemade bags for purchasing	0.616		
14	Avoid using disposable plates, spoons, and glasses for eating/drinking	0.603		
B. Energy				
1	Switch off lights in unoccupied rooms		0.817	
2	Use energy-efficient household appliances		0.756	
3	Open curtains and use natural lights in day time		0.736	
4	Wash laundry by hand whenever possible		0.721	
5	Drying laundry under sunshine (line dry)		0.710	
6	Watch TV programs together with other people in the house		0.694	
7	Reduce ironing (pressing) of clothes		0.675	
8	Turn off the stove whilst doing other work during cooking		0.658	
9	Spend less time on mobile in the day		0.646	
10	Use normal water for washing laundry		0.623	
11	Use normal water whilst taking a bath		0.619	
12	Combine many errands in one trip before motorcycle/car usage		0.609	
C. Water				
1	Avoid keeping the tap running when washing dishes			0.767
2	Turn off the tap whilst cleaning teeth			0.742
3	Wait until having full load of laundry before washing			0.741
4	Water saving by taking short showers			0.727
5	Save water during taking a bath by closing shower while soaping body			0.664
6	Save water during washing one’s face by closing tap while soaping face			0.654
7	Reduce how often face is washed each day			0.642
8	Switch off water pump before overflow			0.616
9	Put only that amount of water in glass that one can drink			0.610

Extraction method: Principal component analysis; rotation method: Varimax with Kaiser normalization; rotation converged in 5 iterations.

**Table 3 ijerph-19-12945-t003:** Determinants of sustainable household consumption.

Explanatory Variables	Coef.	Std. Err.	Marginal Effects		
			Low	Medium	High
Household Dynamics					
Age (Years)	0.001	0.004	−0.001	−0.002	0.003
Gender (1 = Male)	−0.161 **	0.065	0.002	0.050	−0.052
Education (Years)	0.020 **	0.008	−0.004	−0.012	0.016
Residential area (1 = Urban)	0.101 *	0.035	−0.009	−0.020	0.029
Main source of income (1 = Non-agriculture)	0.073	0.073	−0.022	−0.043	0.065
Family size (Numbers)	0.220 *	0.063	0.070	0.101	0.171
Society membership (1 = Yes)	−0.082	0.064	0.026	0.132	−0.158
Internet access (1 = Yes)	−0.177	0.262	0.056	0.072	−0.128
Medium income ^a^	0.032 **	0.015	−0.010	−0.024	0.034
High income ^a^	0.057 **	0.025	0.018	0.043	−0.061
Climate change awareness, importance, and sustainable consumption knowledge					
Climate change awareness (Mean)	0.073 *	0.020	−0.022	−0.026	0.048
Climate change importance (Mean)	0.115 *	0.022	−0.036	−0.087	0.123
Sustainable consumption knowledge (Mean)	0.102 *	0.025	−0.032	−0.045	0.077

LR chi^2^ = 269.123; Prob. > chi^2^ = 0.000; Log likelihood = −1386.954; Pseudo R^2^ = 0.687; * and ** indicate significance level at 1 and 5%, respectively; ^a^ = Base category low-income households.

## Data Availability

The data presented in this study are available on request from the corresponding author.

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
