# Peer review of "Food, Energy, and Water Nexus at Household Level: Do Sustainable Household Consumption Practices Promote Cleaner Environment?"

_ijerph, 2022, doi:10.3390/ijerph191912945_

Round 1
Reviewer 1 Report
The authors have competently and clearly presented the routes of virus spread using the latest literature on Covid-19. They indicate on the basis of the literature that humans are the main link of transmission. They analyse the possible routes of transmission in food chains based on the sources, indicate countermeasures, disinfection and emphasise the role of high temperatures.
A. The authors have been very thorough in presenting the impact of the virus on the domestic environment through commercial activity, access to food and contact with waste. They took great care in documenting the sample selection and size. The chosen method of multivariate statistics excelled in the development of a large dataset.
On the basis of experiments (questionnaires and statistical calculations), authors reached a deep understanding of the influence of gender, education, place of residence, household size and other parameters on their consumption behaviour and waste handling. In addition divided the latter into three categories. It is a pity that the drawing about them is done in three slightly different dimensions, which makes comparisons a bit difficult. I think that this schould be corrected.
Congratulations for developing a household consumption index on a scale of 35 -170 and assigned participants to three groups.
The authors showed some limitations regarding waste disposal practices in particular. I have confidence that in the near future the authors of the reviewed paper, will confront this problem.
I recommend the paper for publication.
Author Response
Please see attached file. Regards.

Reviewer 2 Report
Dear Authors
Apart from some minor grammar and spelling mistakes, you should improve your conclusions, and go further. Improve a little bit the discussion facing the theory.
Author Response
Please see attached file. Regards.

Reviewer 3 Report
The paper appears interesting, the methodology adopted is well known, and the references appear up to date. However, some revisions need to be made to the conclusions before the paper is published.
Conclusions should be well highlighted in a separate paragraph and refer to the literature cited. They should not merely repeat the summary of the work. They must therefore be rewritten
Author Response
Please see attached file. Regards.

Round 2
Reviewer 3 Report
OK!